# COVID-19 reinfection in Liberia: Implication for improving disease surveillance

Godwin E. Akpan[1]*, Luke Bawo[2], Maame Amo-Addae[1], Jallah Kennedy[3]
C. Sanford Wesseh[2], Faith Whesseh[1], Peter Adewuyi[1], Lily Sanvee-Blebo[1],
Joseph Babalola[1], Himiede W. W. Sesay[1], Trokon O. Yeabah[4], Dikena Jackson[2],
Fulton Shannon[2], Chukwuma David Umeokonkwo[1,5], Abraham W. Nyenswah[4],
Jane Macauley[6], Wilhelmina Jallah[7]

1 Division of Field Epidemiology Training Program, African Field Epidemiology Network, Monrovia, Liberia,
2 Department of Planning, Research and Development, Ministry of Health, Monrovia, Liberia, 3 Office of the
Executive Director, Roads To Health (Roads to Rural and Vulnerable Population Health), Galloway, New
Jersey, United States of America, 4 Division of Infectious Disease and Epidemiology, National Public Health
Institute of Liberia, Monrovia, Liberia, 5 Department of Community Medicine, Alex Ekwueme Federal
University Teaching Hospital, Abakaliki, Ebonyi State, Nigeria, 6 Office of the Director General, National
Public Health Institute of Liberia, Monrovia, Liberia, 7 Office of the Minister, Ministry of Health, Monrovia,
Liberia

* profjedidiah@yahoo.com

pone.0265768

University, SINGAPORE

**Data Availability Statement:** All relevant data are
within the paper and its Supporting Information
files.

## Abstract

COVID-19 remains a serious disruption to human health, social, and economic existence.
Reinfection with the virus intensifies fears and raises more questions among countries, with
few documented reports. This study investigated cases of COVID-19 reinfection using
patients' laboratory test results between March 2020 and July 2021 in Liberia. Data obtained
from Liberia's Ministry of Health COVID-19 surveillance was analyzed in Excel 365 and Arc-
GIS Pro 2.8.2. Results showed that with a median interval of 200 days (Range: 99–415), 13
out of 5,459 cases were identified and characterized as reinfection in three counties during
the country's third wave of the outbreak. Eighty-six percent of the COVID-19 reinfection
cases occurred in Montserrado County within high clusters, which accounted for over 80%
of the randomly distributed cases in Liberia. More cases of reinfection occurred among inter-
national travelers within populations with high community transmissions. This study sug-
gests the need for continued public education and surveillance to encourage longer-term
COVID-19 preventive practices even after recovery.

## Introduction

The emergence of coronavirus disease (COVID-19) in December 2019 has affected over 243
million people, with over 4.9 million deaths as of October 24, 2021 [1]. Caused by a novel
severe acute respiratory syndrome coronavirus 2 (SARS-CoV-2), COVID-19 has plagued
every aspect of human life, including the world's economy [2–6]. COVID-19 infection contin-
ues to rise with pockets of reported reinfections across the globe [7].

**Funding:** The author(s) received no specific funding for this work.

**Competing interests:** The authors have declared that no competing interests exist.

The reinfection with SARS-CoV-2 is an emerging public health concern. It raises concerns on how likely and how often it occurs, how soon after the first infection can it happen, how severe is reinfection, who might be at higher risk for reinfection, can it occur after full vaccination, and what it means for a person's immunity [8, 9].

There are different definitions for COVID-19 reinfection. The United States Centers for Disease Control and Prevention (CDC) defines it as a person becoming infected again after full recovery from an initial infection [8]. It has also been defined as individuals infected with different genetic strains of SARS-CoV-2 confirmed by polymerase chain reaction (PCR) and genomic studies [7]. At the same time, some scholars described reinfection with SARS-CoV-2 by having two positive tests separated by a period of greater than 90 days after the initial infection has resolved, as confirmed by two or more consecutive negative tests [10, 11].

Though the rate of reinfection has been reported as rare, many uninvestigated and unreported cases may exist in many countries with ongoing response [7]. Reinfection is more difficult to document when cases of asymptomatic reinfections are involved [7, 12]. Also, the attention of many countries has been on controlling the pandemic and not necessarily looking for cases with possible COVID-19 reinfection. Nonetheless, some countries have reported COVID-19 reinfection in North America, Europe, South America, Asia, and Africa [7, 13–17]. It is essential for countries to continue to monitor the rate and pattern of reinfection in their response to guide policy and management protocol. This study, therefore, set out to identify and characterize COVID-19 reinfection in Liberia using laboratory test results.

## Materials and methods

### Study area

This study was carried out in Liberia using COVID-19 surveillance data between March 2020 and July 2021. Liberia is a West African country with about 5.2 million people [18]. The country is divided into 15 counties and 93 health districts. On July 31, 2021, the country had reported a cumulative total of 5,459 COVID-19 cases with a case fatality rate of 5.0% [19]. Over 80% of these cases occurred in the highly populated Montserrado County [18].

Liberia had experienced three waves of COVID-19 outbreak by the end of July 2021. The first wave was between March 15 and September 28, 2020, with 1,347 cases. The second wave, with 785 cases, started on September 29, 2020, and ended on May 16, 2021. Over 60% (3,327) of the cases occurred during the third wave, which began on May 17, 2021 and was still on as at the time of this report, July 31, 2021.

Through the Incident Management System, the Ministry of Health established some protocols to guide the country's response to the pandemic [20, 21]. These included risk communication to the populace to accept and adopt the non-pharmaceutical public health measures like proper and consistent wearing of face masks, avoiding crowded places, maintaining safe social distance, regular washing of hands and maintaining good respiratory hygiene practices. All incoming international passengers were tested using rapid diagnostic test kits at the airport. All who tested positive were further tested using PCR, and if found positive, they were moved to the designated treatment centers. All contacts of COVID-19 cases, both within the country and international travelers, were traced, followed up, and tested if they met the criteria. International travelers were citizens who visited and returned from other countries or non-citizens who came into the country either by land border crossing, sea, or flight. Mild cases in the community were treated at home using a home-based care approach, while the moderate and severe cases were managed in designated treatment centers.

### Data flow

Testing for COVID-19 during the first and second waves of the pandemic in Liberia was done only at the National Reference Lab (NRL). However, the country does not have the capacity for COVID-19 genetic sequencing and genomics surveillance. Samples were collected from health facilities, counties, and points of entry and sent to the National Reference Lab for PCR test (Fig 1). The sources of the samples were in-coming travelers at the airport, out-going travelers at testing centers, volunteers at walk-through sites and enhanced active surveillance activities, and high-risk contacts in different counties.

However, during the third wave, the number of testing sites has increased with the introduction of rapid diagnostic tests (RDTs), and some private laboratories and organizations have acquired the capacity and license to conduct COVID-19 PCR tests.

The COVID-19 surveillance data in Liberia is managed online through DATAPOINT, an online data management system designated by the Incident Management System (IMS). As of July 2021, all the COVID-19 test results and accompanying data were submitted to MOH/NPHIL through the DATAPOINT. All service points with the capacity for PCR (e.g., private labs) collect samples, test, and submit their results through the DATAPOINT. Those with the capacity for RDT (e.g., airport and other points of entry, mobile labs, health facilities, treatment units, counties, and enhanced active surveillance sites) do the same. However, those

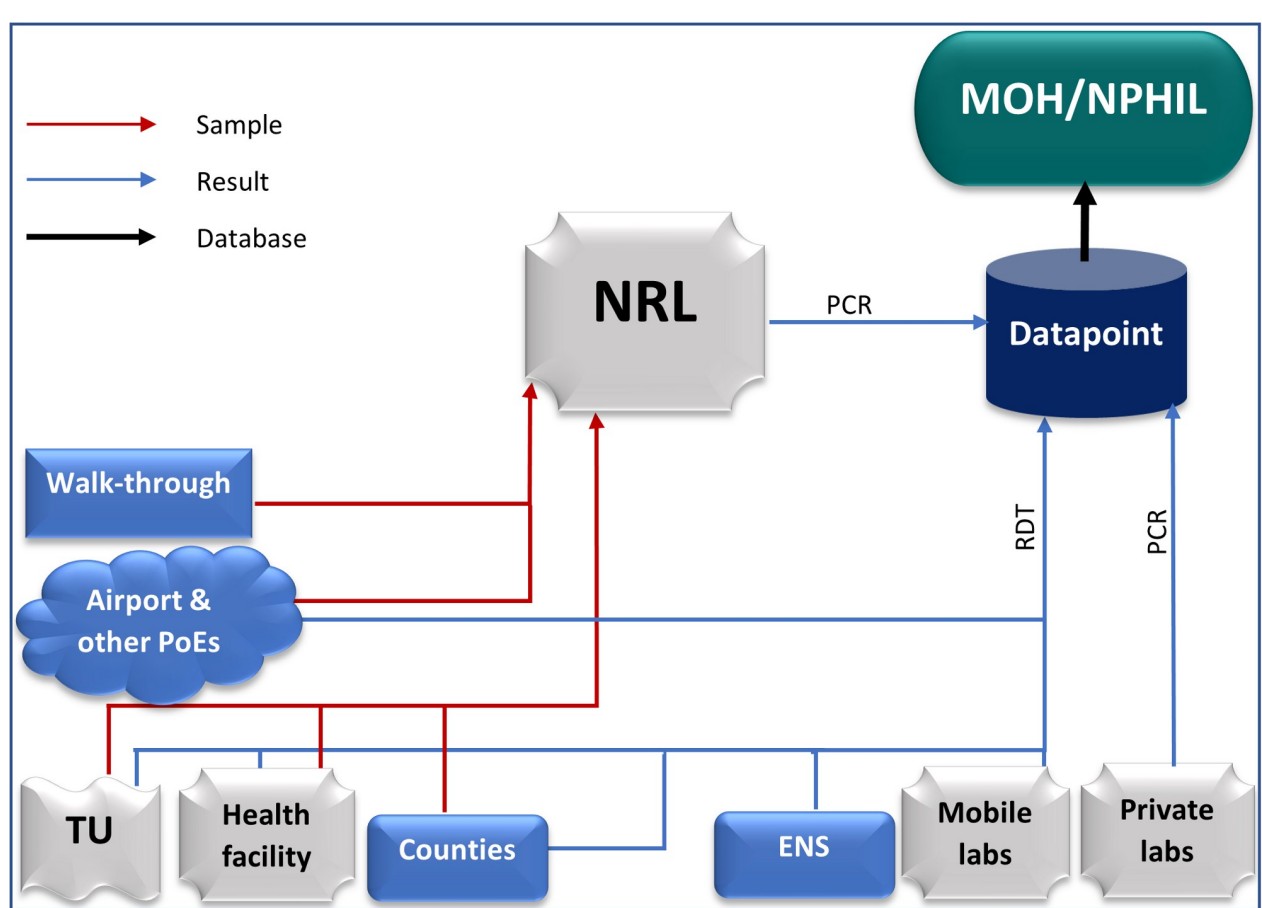

**Fig 1. COVID-19 Epi-data flowchart in Liberia, July 2021. Note:** Ministry of Health (MOH), National Public Health Institute of Liberia (NPHIL), National Reference Lab (NRL), Treatment Unit (TU), PoE (Points of Entry), and Enhanced Active Surveillance (ENS).

sample collection sites that cannot test either with PCR or RDT (e.g., walk-through sites) collect samples and send them to the NRL for testing and onward submission to the MOH/ NPHIL through the datapoint (Fig 1).

All the COVID-19 data are consolidated, cleaned, and validated in the DATAPOINT. The data is then shared with the MOH/NPHIL for analysis and interpretation for decision-making and subsequent archiving.

## Data source and variables of interest

The data for this study was obtained from the Ministry of Health COVID-19 surveillance data on DATAPOINT. The variables of interest were case ID, name, sex, age, address, phone number, date of sample collection, and test result (S1 File).

## COVID-19 reinfection case definition

Due to lack of capacity for genomic sequencing, a case was defined as reinfection when they met the following criteria: (1) recorded another positive result after an interval equal to or greater than 90 days from initial laboratory confirmed infection, and (2) had at least one negative PCR result in between the two infections [10]. The date interval for the reinfection was calculated from the date of the first negative PCR results after the initial infection to the date of the next positive result [10].

Due to unique identification (Case ID) challenges, we concatenated the patient's name, age, phone number, and address as a proxy for a unique identification number.

## Data analysis

The cases of COVID-19 reinfection were identified using the case definition. The basic demographics were analyzed in Microsoft Excel 365 and presented in tables and charts.

The geospatial distribution of the cases across counties and communities was analyzed using ArcGIS Pro 2.8.2. Spatial locations of daily COVID-19 cases were digitized, classified, and presented in maps. The spatial pattern of distribution of cases was assessed using the spatial autocorrection described below.

## Assessment of spatial pattern in the distribution of COVID-19 cases in Liberia

To assess the overall distribution pattern of COVID-19 cases across the country–whether clustered, dispersed, or random, we used Spatial Autocorrelation (Global Moran's I). We used Cluster and Outlier Analysis (Anselin Local Moran's I) to identify the specific area and how the pattern exists. A cluster in this case is defined as a contiguous region of high case density, separated from other such clusters by contiguous regions of low case density [22]. With the null hypothesis that the pattern is random or that no spatial autocorrelation is present in the transmission or distribution of the COVID-19 cases, we applied Global Moran's I in three different Input Feature Classes: county, district, and the location of individual cases (point). In the same vein, we applied Local Moran's I in two different Input Feature Classes: district and point.

To ascertain the influence of COVID-19 cases on nearby neighboring or distant cases (target feature), we selected an Inverse Distance Conceptualization of Spatial Relationships with Euclidean Distance method in the Global Moran's I tool. We also set the same parameters as in Global Moran's I for Local Moran's I, with Row standardization and 999 Permutations to improve the precision of the pseudo p-value and the random distribution of cases.

## Ethical consideration

Ethics approval was obtained from the Ministry of Health through the COVID-19 Incident Management System (IMS) leadership for COVID-19 Pandemic Response in Liberia (S2 File). The IMS comprises the leadership of the Ministry of Health and the National Public Health Institute of Liberia. The secondary data was produced as part of the routine activities by the authors who were members and technical partners of epidemiology unit of the IMS. Written informed consent was obtained from the cases. The patient personal identifiable information was handled with the utmost confidentiality. The dataset was stored in a password-protected computer. The personal identifiers were scrambled after the initial identification of the reinfection cases using the proxy parameters.

## Results

As of the end of July 2021, 5,459 cases of COVID-19 have been reported in Liberia. Of the confirmed cases, 0.2% (13) were identified to have been reinfected with COVID-19. The median interval between initial infection and reinfection was 200 days (Range: 99–415). Of the 13 cases, 62% (8) were males. The age range of the cases was 21–74 years with a modal age of 58. Seventy-seven percent (10) of the cases were international travelers (Table 1).

COVID-19 reinfections were reported in three out of the 15 counties in Liberia (Fig 2). Montserrado County accounted for 84.6% (11) of the cases (Fig 2). These 11 cases were reported in five out of seven districts in the county (Fig 3). The other two cases were reported in Bomi and Grand Bassa counties (Figs 2 and 3).

The general distribution pattern of COVID-19 cases in Liberia appeared to be random across the country according to Global Moran's I index of -0.075307 and z-score of -0.101480 with a p-value of 0.919170 at the county level, and a Moran's Index of -0.000246 and z-score of -1.339181 with a p-value of 0.180512 at individual cases location level (Table 2 and S1 Fig). However, the math for the autocorrelation statistic (z-score) cannot accurately be solved with an unaggregated incident where all input values are 1, as in the case of cases at point level. The aggregated COVID-19 cases data at the district level with a z-score of 7.334529 and p-value of 0.000000 indicated that there is a less than 1% likelihood that any clustered pattern could be the result of random chance (Table 2 and S1 Fig). Local Moran's I output for both aggregated and unaggregated data agree with the general random distribution of cases across the country

**Table 1. The distribution of COVID-19 reinfection cases in Liberia, March 2020 –July 2021.**

| Case | County | Age (years) | Sex | Infections Interval (Days) | International Traveler |
|------|--------|-------------|-----|----------------------------|------------------------|
| C1 | Bomi | 47 | Female | 178 | Yes |
| C2 | Grand Bassa | 35 | Male | 410 | Yes |
| C3 | Montserrado | 58 | Female | 99 | Yes |
| C4 | Montserrado | 56 | Male | 386 | No |
| C5 | Montserrado | 29 | Male | 99 | Yes |
| C6 | Montserrado | 42 | Male | 181 | Yes |
| C7 | Montserrado | 58 | Female | 415 | No |
| C8 | Montserrado | 30 | Female | 325 | Yes |
| C9 | Montserrado | 33 | Male | 200 | Yes |
| C10 | Montserrado | 51 | Male | 182 | Yes |
| C11 | Montserrado | 58 | Male | 320 | Yes |
| C12 | Montserrado | 21 | Female | 231 | No |
| C13 | Montserrado | 74 | Male | 155 | Yes |

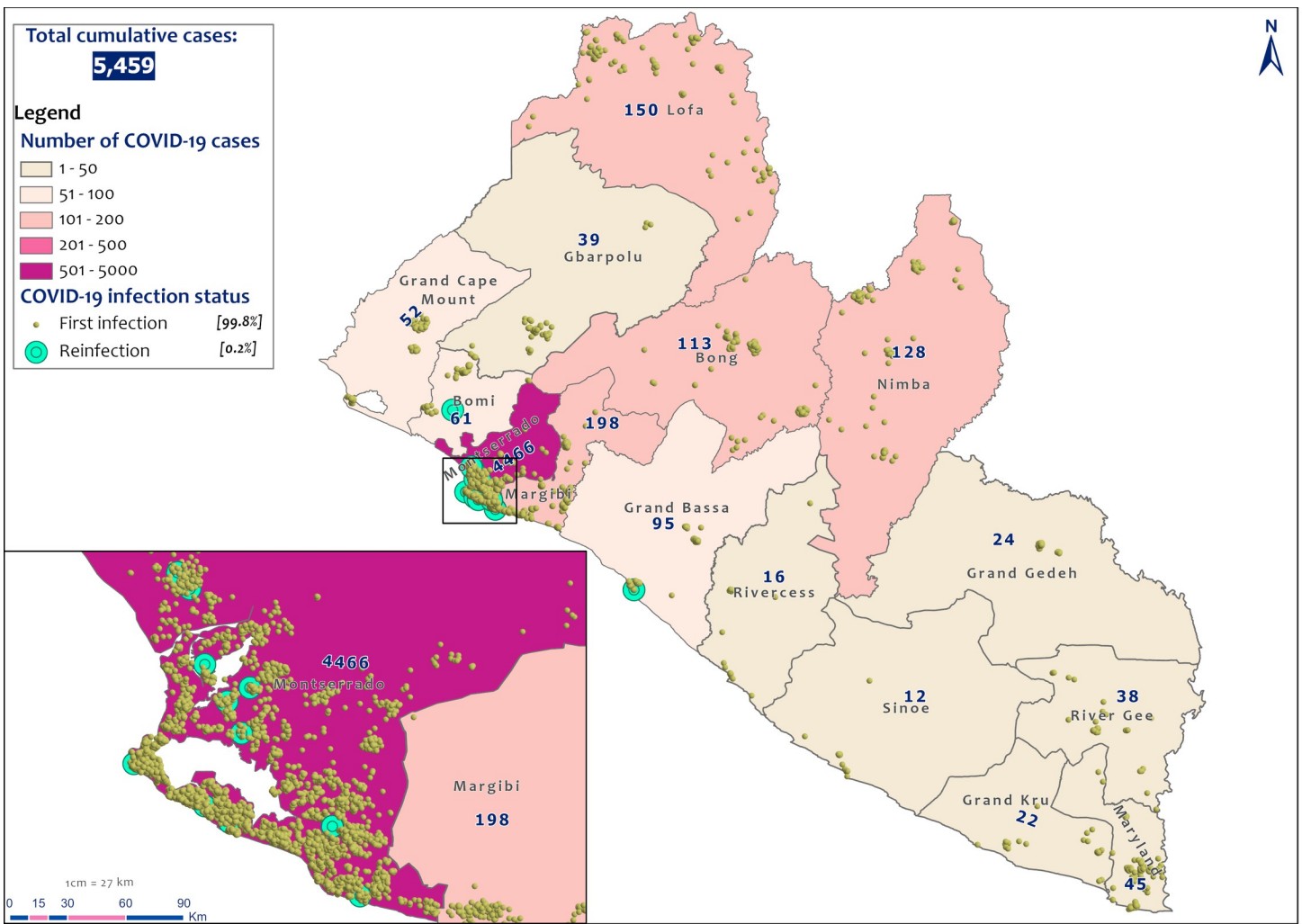

**Fig 2. Spatial distribution of COVID-19 cases by county in Liberia, March 15, 2020 to July 31, 2021.** Data source: The Incident Management System for COVID-19 Response in Liberia (S2 File).

with possible autocorrelation among clustered cases. It revealed sporadic low clusters (density) of cases across the country and high clusters within the epicenter Montserrado County, as well as Bomi, and Maryland counties (Figs 4 and 5 and S1 File). In Liberia, reinfection occurred more in communities with high transmission rates within high clusters (Fig 5). Like Montserrado County, the reinfection case in Bomi County occurred within a high cluster environment with similar features in a close neighborhood (12.82) to Montserrado (Fig 4).

All initial infections occurred during the first and second waves of COVID-19 in Liberia. Five occurred during the first wave and eight during the second wave (Fig 6). All the reinfections occurred during the country's third wave of COVID-19 (Fig 6).

## Discussion

Cases of COVID-19 reinfections were identified in three counties in Liberia using histories of patients' laboratory tests and results [10]. Analysis showed that these cases were mainly international travelers who may have been exposed to different strains of the virus. This information can be used in public education to encourage continued COVID-19 preventive practices

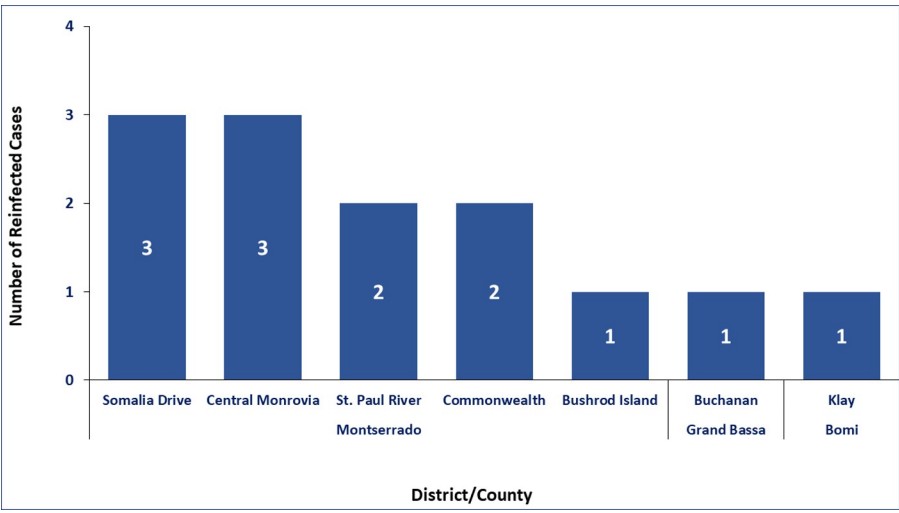

**Fig 3. Distribution of reinfection by district/county in Liberia, March 15, 2020 to July 31, 2021.**

even after recovering from an initial infection or being fully vaccinated [23]. Although vaccination had started in Liberia, it was not captured in DATAPOINT, so the vaccination status of the reinfection cases could not be determined. The low proportion of reinfection rate among confirmed cases of COVID-19 was in keeping with earlier reports which reported a prevalence of less than one percent [7, 12]. The proportion could be more with the emergence of new variants of the virus.

Reinfections may have been due to high community transmissions, as indicated in our analysis, especially in the densely populated urban Montserrado, where two-thirds of all cases in the country occurred [19]. Siraj et al. [24] observed that COVID-19 infections increase in urban populations as more people interact with less or no observation of protection measures. However, the observation of protection measures was not measured in this study. The non-occurrence of reinfection in about 80% of counties may have depended on the type of populations, society settings, lifestyle, and travel rather than the distribution pattern [25]. This result may have been an indication that it is unlikely for autocorrelation to exist among COVID-19 cases across a large area with dissimilar environmental, social, physical, and biological (medical) features. This may also explain the unlikeliness for COVID-19 reinfection to become pervasive in a region or state with different human and environmental features.

Reinfections may have occurred due to zero or waned natural immunity within an infection interval of more than three months to a year and two months [26–28]. However, the COVID-

**Table 2. Assessment of spatial pattern in the distribution of COVID-19 cases in Liberia, March 15, 2020 to July 31, 2021.**

| Spatial Autocorrelation | County | District | Point |
|---|---|---|---|
| Moran's Index | -0.075307 | 0.436056 | -0.000246 |
| Expected Index | -0.071429 | -0.010989 | -0.000182 |
| Variance | 0.001461 | 0.003715 | 0.000000 |
| z-score | -0.101480 | 7.334529 | -1.339181 |
| p-value | 0.919170 | 0.000000 | 0.180512 |
| Inference | Given the z-score of -0.10148, the pattern does not appear to be significantly different than random | Given the z-score of 7.334529, there is a less than 1% likelihood that this clustered pattern could be the result of random chance | Given the z-score of -1.339181, the pattern does not appear to be significantly different than random |

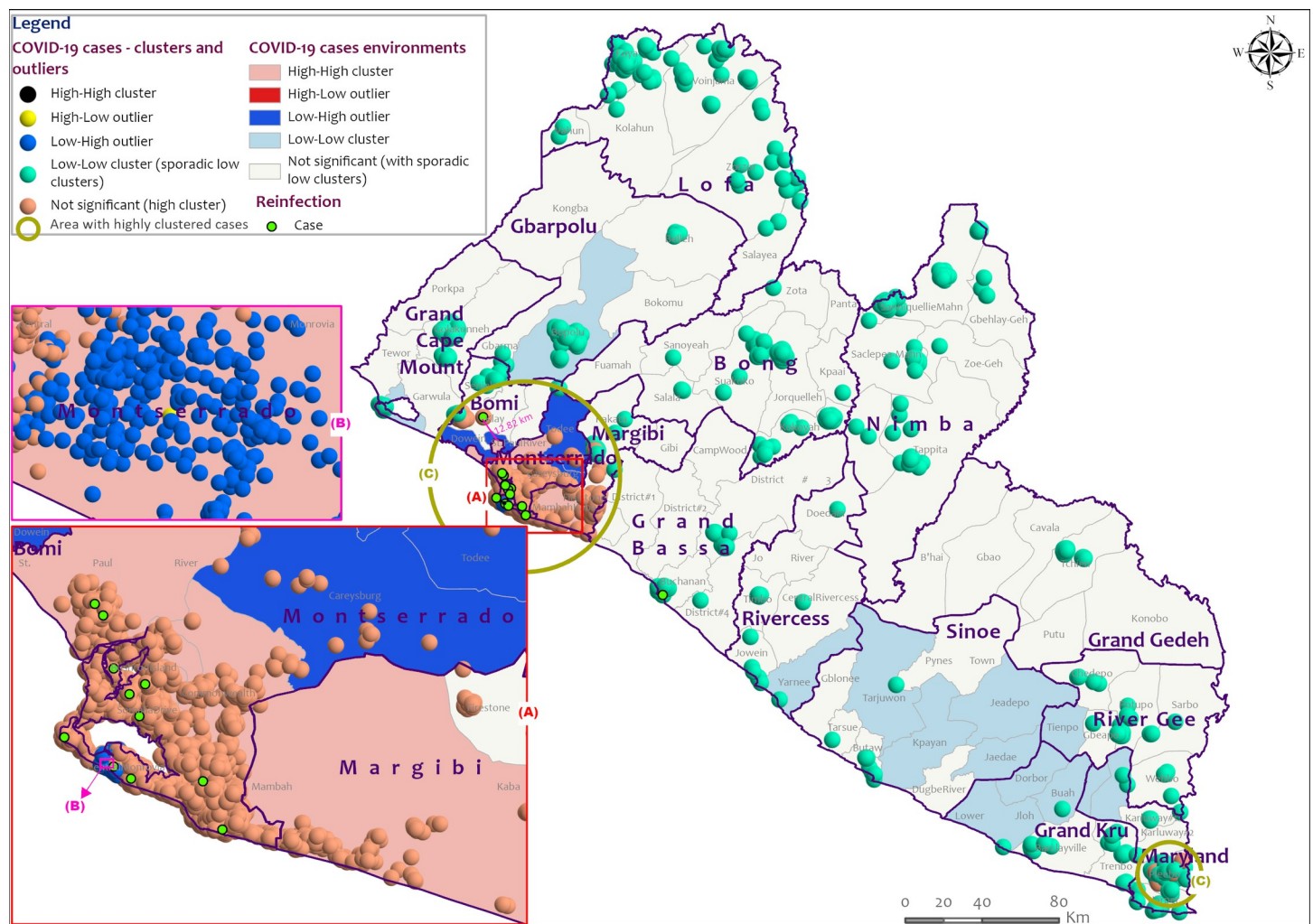

**Fig 4. Distribution pattern of COVID-19 cases in Liberia, March 15, 2020 to July 31, 2021.** Data source: The Incident Management System for COVID-19 Response in Liberia (S2 File).

19 protective immunity duration is yet to be clearly defined [28–31]. Also, advancing age may not have been responsible since all the cases (except one) were below 60 years. It has been established that "protection against reinfection is lower in individuals aged 65 years and older" [28].

In any case, all the reinfections were recorded in the third wave of the COVID-19 outbreak, where the Delta variant was the predominant strain that may have been responsible for the reinfections [32–34]. However, this cannot be confirmed due to the lack of genomic sequencing for each case [7, 35]. Availability of laboratory capacity for genomic sequencing could have helped establish the variants involved in the reinfection and the reasons for the changing dynamics of the pandemics in the population. In Liberia, genomic sequencing will save time and costs and increase real-time identification of disease causative agents, especially for pathogens that require a shorter turnaround time. Real-time diagnosis would improve prompt disease management and health outcome.

While we may have established the occurrence of reinfection, we cannot describe and compare the two infections' clinical features [9, 15]. This is due to the absence of linkage between clinical and epi-surveillance data in the national COVID-19 database. This calls for a well-

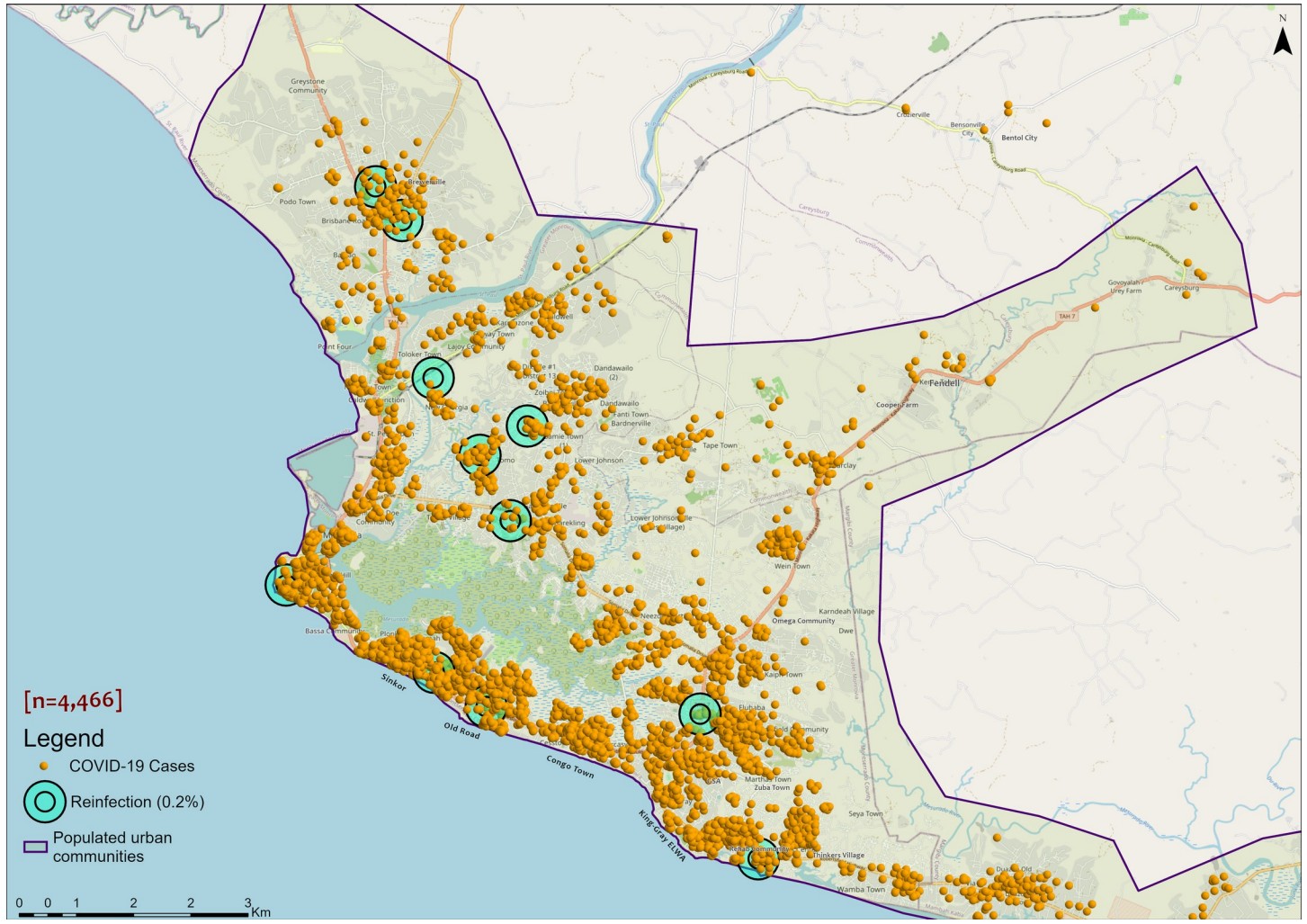

**Fig 5. Distribution of COVID-19 cases in Montserrado communities, Liberia, March 15, 2020 to July 31, 2021.** Data source: The Incident Management System for COVID-19 Response in Liberia (S2 File).

established linkage of the national epidemiological data with the clinical database and the urgent need to develop capacity for and commence COVID-19 genomic surveillance.

The absence or limited use of surveillance can have public health implications because it could be likely that reinfection with COVID-19 and other infectious diseases could spur other waves. It is imperative to improve disease surveillance and management, ranging from formulating robust alert mechanisms, case investigation, timely sample testing, and dissemination of results to contact tracing, strategically equitable resource allocation, and, when necessary, early treatment. In weak, impoverished health systems such as Liberia, robust public health disease surveillance could help to reduce the high burden of diseases and the cost of managing outbreaks. Additionally, the benefits of improving disease surveillance can lessen the impacts of outbreaks on routine health services.

## Conclusion

Of the 5,459 recorded COVID-19 cases in Liberia as of July 31, 2021, 13 were reinfections. Within the randomly distributed cases of COVID-19 with sporadic low clusters across the

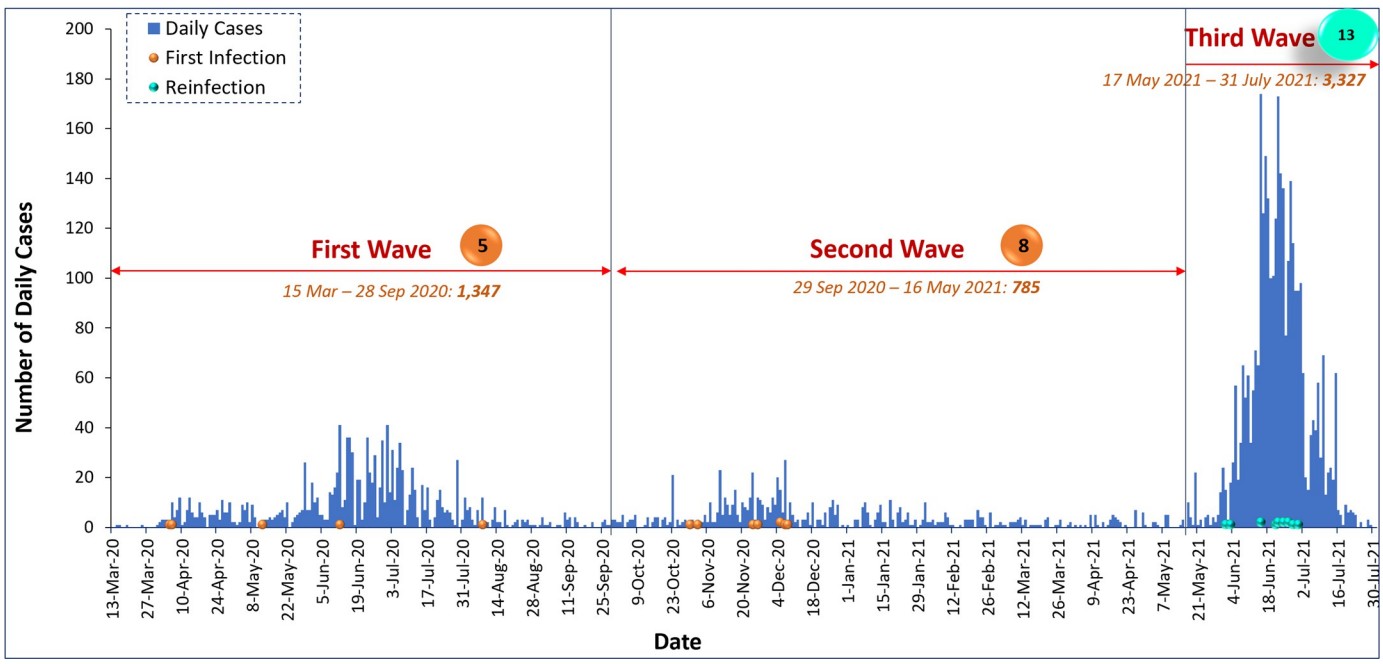

**Fig 6. COVID-19 Epi-curve with possible reinfection in Liberia, March 2020 –July 2021.**

country, reinfections were predominately in densely populated areas with high clusters (case density) in three of the 15 counties. Sixty-two percent of the reinfected cases were males, and 77% were international travelers. Our findings confirmed that COVID-19 reinfection is occurring in Liberia and calls for the need to maintain surveillance to gather vital information for policy development and response activities. There is a need for long-term transmission mitigation efforts to further prevent another outbreak and impact on routine health services. The government and her partners need to urgently develop capacity for and deploy systems for genomic and other surveillance programs, not just for COVID-19 alone but all the infectious diseases.

## Supporting information

**S1 File. A file containing COVID-19 daily and reinfection cases in Liberia, March 15, 2020 to July 31, 2021.**
(XLSX)

**S2 File. Ethics approval from the Ministry of Health through the COVID-19 Incident Management System (IMS) leadership for COVID-19 Pandemic Response in Liberia.**
(PDF)

**S1 Fig. Assessment of spatial pattern in the distribution of COVID-19 case in Liberia, March 15, 2020 to July 31, 2021.**
(TIF)

## Author Contributions

**Conceptualization:** Godwin E. Akpan, Maame Amo-Addae, Jallah Kennedy.

**Data curation:** Godwin E. Akpan, Luke Bawo, Jallah Kennedy, Trokon O. Yeabah, Dikena Jackson, Fulton Shannon.

**Formal analysis:** Godwin E. Akpan, Luke Bawo, Jallah Kennedy, Peter Adewuyi, Joseph Babalola, Chukwuma David Umeokonkwo, Abraham W. Nyenswah.

**Investigation:** Godwin E. Akpan, Luke Bawo, C. Sanford Wesseh, Faith Whesseh, Peter Adewuyi, Lily Sanvee-Blebo, Himiede W. W. Sesay, Trokon O. Yeabah, Dikena Jackson, Fulton Shannon, Abraham W. Nyenswah, Jane Macauley, Wilhelmina Jallah.

**Methodology:** Godwin E. Akpan, Luke Bawo, Maame Amo-Addae, Jallah Kennedy, C. Sanford Wesseh, Faith Whesseh, Peter Adewuyi, Lily Sanvee-Blebo, Joseph Babalola, Himiede W. W. Sesay, Trokon O. Yeabah, Dikena Jackson, Fulton Shannon, Chukwuma David Umeokonkwo, Abraham W. Nyenswah, Jane Macauley, Wilhelmina Jallah.

**Project administration:** Maame Amo-Addae.

**Resources:** Godwin E. Akpan, Maame Amo-Addae, Chukwuma David Umeokonkwo, Jane Macauley.

**Software:** Godwin E. Akpan.

**Supervision:** Maame Amo-Addae, Wilhelmina Jallah.

**Validation:** Godwin E. Akpan, Luke Bawo, Jallah Kennedy, C. Sanford Wesseh, Trokon O. Yeabah, Dikena Jackson, Fulton Shannon.

**Visualization:** Godwin E. Akpan.

**Writing – original draft:** Godwin E. Akpan, Maame Amo-Addae, Jallah Kennedy, Faith Whesseh, Peter Adewuyi, Lily Sanvee-Blebo, Joseph Babalola, Himiede W. W. Sesay, Chukwuma David Umeokonkwo.

**Writing – review & editing:** Godwin E. Akpan, Luke Bawo, Maame Amo-Addae, Jallah Kennedy, C. Sanford Wesseh, Faith Whesseh, Peter Adewuyi, Lily Sanvee-Blebo, Joseph Babalola, Himiede W. W. Sesay, Trokon O. Yeabah, Dikena Jackson, Fulton Shannon, Chukwuma David Umeokonkwo, Abraham W. Nyenswah, Jane Macauley, Wilhelmina Jallah.

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
