## [Decision Letter · Decision Letter 0]

11 Jan 2022

PONE-D-21-36066COVID-19 reinfection in Liberia: implication for improving disease surveillancePLOS ONE

Dear Dr. Akpan,

Thank you for submitting your manuscript to PLOS ONE. After careful consideration, we feel that it has merit but does not fully meet PLOS ONE’s publication criteria as it currently stands. Therefore, we invite you to submit a revised version of the manuscript that addresses the points raised during the review process. Specifically, Reviewer 1 questioned how generalizable the results from a single county can be, and requested for an analysis of the spatial clustering of cases. Please address these, and other comments from the reviewers.

We look forward to receiving your revised manuscript.

Kind regards,

Siew Ann Cheong, Ph.D.

Academic Editor

PLOS ONE

Journal Requirements:

a) Did participants provide their written or verbal informed consent to participate in this study?

3. We note that Figures 2 and 4 in your submission contain map/satellite images which may be copyrighted. All PLOS content is published under the Creative Commons Attribution License (CC BY 4.0), which means that the manuscript, images, and Supporting Information files will be freely available online, and any third party is permitted to access, download, copy, distribute, and use these materials in any way, even commercially, with proper attribution. For these reasons, we cannot publish previously copyrighted maps or satellite images created using proprietary data, such as Google software (Google Maps, Street View, and Earth). For more information, see our copyright guidelines: http://journals.plos.org/plosone/s/licenses-and-copyright.

a) You may seek permission from the original copyright holder of Figures 2 and 4 to publish the content specifically under the CC BY 4.0 license.  

Reviewers' comments:

Reviewer's Responses to Questions

**Comments to the Author**

1. Is the manuscript technically sound, and do the data support the conclusions?

Reviewer #1: No

Reviewer #2: Yes

2. Has the statistical analysis been performed appropriately and rigorously? 

Reviewer #1: No

Reviewer #2: Yes

3. Have the authors made all data underlying the findings in their manuscript fully available?

Reviewer #1: Yes

Reviewer #2: Yes

4. Is the manuscript presented in an intelligible fashion and written in standard English?

Reviewer #1: Yes

Reviewer #2: Yes

5. Review Comments to the Author

Reviewer #1: As mentioned, more than 80% cases occurred in one county and 86% cases of reinfection from the same county. The generalizability of this data for the whole country is questionable.

As data on geolocation was available, Moran’s Index should have been calculated - clustering for events?

Majority of reinfection cases were international travelers - How was international travel defined?

Details related to in country measures to mitigate spread of COVID-19 will aid in better understanding of the scenario.

Reviewer #2: Seems like an interesting article. Few suggestions:

1. PAGE 8 - LINE 11 - we can remove the word "got sick' in bracket.

2. In discussion, kindly add few points on the necessity of having Advanced labs in developing countries to get the genome sequencing done so that we can easily assess who will respond to which Cocktail antibody therapy.

6. PLOS authors have the option to publish the peer review history of their article (what does this mean?). If published, this will include your full peer review and any attached files.

Reviewer #1: **Yes: **Mohan Kumar

Reviewer #2: **Yes: **Dr Sayak Roy

---

## [Author Response · Author response to Decision Letter 0]

16 Feb 2022

Editor's comment:

1. Specifically, Reviewer 1 questioned how generalizable the results from a single county can be, and requested for an analysis of the spatial clustering of cases. Please address these, and other comments from the reviewers

Response:

Thank you for the thorough review of our manuscript and for useful suggestions to improve the understanding of the work. We have carefully gone through the comments and made modifications as suggested in the work. We have also responded to the comments here for ease of reference to the changes made in the work. We hope that the work is now acceptable for publication and will be available to respond to further clarification, if any.

Journal Requirements:

1. Please ensure that your manuscript meets PLOS ONE's style requirements, including those for file naming. The PLOS ONE style templates can be found at https://journals.plos.org/plosone/s/file?id=wjVg/PLOSOne_formatting_sample_main_body.pdf and https://journals.plos.org/plosone/s/file?id=ba62/PLOSOne_formatting_sample_title_authors_affiliations.pdf.

Response:

Thank you for the support link. We have used the attached link to format the manuscript to meet PLOS ONE’s style requirements.

a) Did participants provide their written or verbal informed consent to participate in this study?

b) If consent was verbal, please explain 

i) why written consent was not obtained

ii) how you documented participant consent, and 

iii) whether the ethics committees/IRB approved this consent procedure.

Response:

Thank you for pointing out this. Written informed consent was obtained from the cases. This has been updated in the ethical consideration section of the work. [Lines 160 and 161]

3. We note that Figures 2 and 4 in your submission contain map/satellite images which may be copyrighted. All PLOS content is published under the Creative Commons Attribution License (CC BY 4.0), which means that the manuscript, images, and Supporting Information files will be freely available online, and any third party is permitted to access, download, copy, distribute, and use these materials in any way, even commercially, with proper attribution. For these reasons, we cannot publish previously copyrighted maps or satellite images created using proprietary data, such as Google software (Google Maps, Street View, and Earth). 

We require you to either (1) present written permission from the copyright holder to publish these figures specifically under the CC BY 4.0 license, or (2) remove the figures from your submission 

Response:

Figures 2 and 4 were maps created by the authors for the work using ArcGIS Pro Version 2.8.2. They are not copyrighted materials, and so they do not need copyright permission. They can be published under the creative commons attribution license (CC BY 4.0).

Reviewer 1:

1. As mentioned, more than 80% cases occurred in one county and 86% cases of reinfection from the same county. The generalizability of this data for the whole country is questionable.

Response:

Thank you for these points. The authors did not set out to make generalizations but rather describe the patterns of reinfection observed in the routine surveillance data and make a case for providing laboratory capacity for genomic sequencing in the country. To address the concern raised, the authors have added a section to assess the spatial pattern of distribution of the cases [Lines 130-145]. 

It is also important to note that: 

1. Montserrado County accounts for the majority of the population of Liberia's 5.2 million. Of the 3,476,608 population of Liberia based on the 2008 census, Montserrado County made up 1,118,241 (32.2%) of the 15 counties. https://www.lisgis.net/page_info.php?7d5f44532cbfc489b8db9e12e44eb820=MzQy

2. Considering the high rates (77%) of reinfection among [international travelers, the hub of travel is Montserrado with the only two international airports. Additionally, the dense population and high community transmission and traditional family and friends contacts in Montserrado means that those travelers also made contacts with others in Montserrado County

3. The authors have analyzed several data points and provided reports to key stakeholders that have shown that Montserrado County data can be generalizable to a greater extent to the rest of Liberia on several attributes, including testing, cases, contacts, hospitalization. 

2. As data on geolocation was available, Moran’s Index should have been calculated - clustering for events?

Response:

An additional section to address this has been added, and it reads, 

“To assess the overall distribution pattern of COVID-19 cases across the country – whether clustered, dispersed, or random, we used Spatial Autocorrelation (Global Moran's I). We used Cluster and Outlier Analysis (Anselin Local Moran's I) to identify the specific area and how the pattern exists. With the null hypothesis that the pattern is random or that no spatial autocorrelation is present in the transmission or distribution of the COVID-19 cases, we applied Global Moran's I in three different Input Feature Classes: county, district, and the location of individual cases (point). In the same vein, we applied Local Moran's I in two different Input Feature Classes: district and point.

To ascertain the influence of COVID-19 cases on nearby neighboring or distant cases (target feature), we selected an Inverse Distance Conceptualization of Spatial Relationships with Euclidean Distance method in the Global Moran's I tool. We also set the same parameters as in Global Moran's I for Local Moran's I, with Row standardization and 999 Permutations to improve the precision of the pseudo p-value and the random distribution of cases.” The results of the analysis have also been added to the work. [Lines 186-201]

3. Majority of reinfection cases were international travelers - How was international travel defined?

Response:

The authors defined international travelers as “citizens who visited and returned from other countries or non-citizens who came into the country either by land border crossing, sea or flight.” This has been added to the work. [Lines 80-82]

Review 2:

1. PAGE 8 - LINE 11 - we can remove the word "got sick' in bracket

Response:

The correction has been effected as suggested. [Line 44]

2. In discussion, kindly add few points on the necessity of having Advanced labs in developing countries to get the genome sequencing done so that we can easily assess who will respond to which Cocktail antibody therapy.

Response:

Thank you for this suggestion. The authors have added a few more comments to buttress the need for genomic sequencing. [Lines 249-253]

---

## [Decision Letter · Decision Letter 1]

8 Mar 2022

COVID-19 reinfection in Liberia: implication for improving disease surveillance

PONE-D-21-36066R1

Dear Dr. Akpan,

We’re pleased to inform you that your manuscript has been judged scientifically suitable for publication and will be formally accepted for publication once it meets all outstanding technical requirements.

Kind regards,

Siew Ann Cheong, Ph.D.

Academic Editor

PLOS ONE

Additional Editor Comments (optional):

Reviewers' comments:

Reviewer's Responses to Questions

**Comments to the Author**

1. If the authors have adequately addressed your comments raised in a previous round of review and you feel that this manuscript is now acceptable for publication, you may indicate that here to bypass the “Comments to the Author” section, enter your conflict of interest statement in the “Confidential to Editor” section, and submit your "Accept" recommendation.

Reviewer #1: All comments have been addressed

2. Is the manuscript technically sound, and do the data support the conclusions?

Reviewer #1: Yes

3. Has the statistical analysis been performed appropriately and rigorously? 

Reviewer #1: Yes

4. Have the authors made all data underlying the findings in their manuscript fully available?

Reviewer #1: Yes

5. Is the manuscript presented in an intelligible fashion and written in standard English?

Reviewer #1: Yes

6. Review Comments to the Author

Reviewer #1: (No Response)

7. PLOS authors have the option to publish the peer review history of their article (what does this mean?). If published, this will include your full peer review and any attached files.

Reviewer #1: **Yes: **Mohan Kumar

---

## [Editor Report · Acceptance letter]

15 Mar 2022

PONE-D-21-36066R1 

COVID-19 Reinfection in Liberia: Implication for Improving Disease Surveillance 

Dear Dr. Akpan:

I'm pleased to inform you that your manuscript has been deemed suitable for publication in PLOS ONE. Congratulations! Your manuscript is now with our production department. 

Kind regards, 

on behalf of

Dr. Siew Ann Cheong 

Academic Editor

PLOS ONE